# Effectiveness of integrating cervical cancer prevention strategies into HIV care programmes: A mixed-methods systematic review protocol

Kimeshnee Govindsamy[1,2]*, Susanne Noll[2], Ntombifuthi Blose[3], Edina Amponsah-Dacosta[2]

**1** Division of Epidemiology and Biostatistics, School of Public Health and Family Medicine, University of Cape Town, Western Cape, South Africa, **2** Vaccines for Africa Initiative (VACFA), Faculty of Health Sciences, School of Public Health and Family Medicine, University of Cape Town, Cape Town, South Africa, **3** Health Systems Trust, Durban, South Africa

* GVNKIM019@myuct.ac.za

## Abstract

### Introduction

Cervical cancer, which is the fourth most frequently diagnosed cancer among women globally, remains a significant health burden despite being preventable and treatable, exposing gaps in accessing prevention and control services. Adolescent girls and young women (AGYW) living with HIV face heightened risk of persistent HPV infection, a primary cause of cervical cancer, making this population the ideal target for preventing cervical cancer before HPV exposure or disease progression. The overlap of cervical cancer and HIV exacerbates public health challenges, urging intensified efforts in bolstering prevention and control measures. Integration of cervical cancer prevention strategies into HIV care programs shows promise in effectively addressing this dual burden.

### Methods

To evaluate the effectiveness of integrating cervical cancer prevention strategies within HIV care programs, a mixed-methods systematic review will be conducted. A comprehensive Boolean search for literature published and indexed in PubMed, Cochrane Library, EBSCO Host, Web of Science, Scopus, and Google Scholar will be conducted, without imposing any language restrictions. This review will be conducted in alignment with the Joanna Briggs guidelines on systematic reviews together with the Preferred Reporting Items for Systematic Reviews and Meta-Analyses (PRISMA) guidelines. Data from eligible studies will be extracted and synthesized, and their quality assessed.

**Data availability statement:** No datasets will be generated or analysed for this systematic review protocol. All relevant data from this study will be made available upon study completion.

**Funding:** The authors have declared that no financial support will be required for this systematic review.

**Competing interests:** The authors have declared that no competing interests exist.

## Discussion

There is limited understanding of the effectiveness of integrating cervical cancer prevention and HIV care in the real-world setting. While some studies touch on integration, focus tends to be on cervical cancer screening alone, neglecting vaccination, treatment of precancerous lesions, and education programs. Previous reviews on this focus are outdated, surpassing six years. This systematic review aims to fill these evidence gaps by thoroughly evaluating the challenges and opportunities associated with integrating the full complement of HPV prevention strategies and HIV care programs. The anticipated findings could enhance service delivery models aimed at reducing cervical cancer incidence and mortality among AGYW living with HIV.

### Trial registration

Systematic review registration: PROSPERO registration number: CRD42024535821.

## Introduction

### Background

Cervical cancer ranks as the fourth most frequently diagnosed cancer among women globally, with current global estimates from 2022 indicating that 660 000 women are diagnosed with cervical cancer and 350 000 women die from this disease annually, according to the World Health Organization (WHO) [1]. Persistent infection with the Human Papillomavirus (HPV) following sexual transmission is the primary sufficient cause of cervical cancer [2]. Furthermore, cervical cancer is the most common HPV-related disease [2]. Most HPV infections resolve spontaneously and do not cause symptomatic disease. However, persistent infection with specific high-risk HPV types (most frequently HPV types 16 and 18) may lead to pre-cancerous lesions and if untreated these lesions may progress to invasive cervical cancer [2]. According to Okunade [2] approximately 99.7% of cervical cancer cases are caused by persistent high-risk HPV infection. HPV is estimated to infect approximately 291 million women globally, with a significantly higher prevalence among women under the age of 25 years [3].

Although HPV is the underlying sufficient cause of cervical cancer there are other risk factors associated with this disease which include smoking, increased parity, long-term use of oral contraceptives as well as infection with human immunodeficiency virus (HIV) [2]. Studies have shown that women who are HIV-positive face a six-fold increased risk of developing cervical cancer compared to those without HIV, which further exacerbates the burden of cervical cancer [4]. This heightened risk stems from a multifaceted interplay of biological and societal factors. Among these are the direct impact of HIV on the immune regulation of HPV, accelerated disease advancement in HIV-positive women, extended life expectancy due to antiretroviral therapy, and obstacles such as stigma, poverty, and gender-related barriers that hinder women from accessing timely care [5]. Compelling evidence indicates that women infected with HIV face an elevated risk of persistent infection with multiple

types of HPV at an early age, specifically between the ages of 13 and 18 years, which contributes to an increased likelihood of developing cervical cancer at a younger age [3]. This dual burden of HPV and HIV poses a complex challenge to healthcare systems worldwide, especially among AGYW living with HIV.

Cervical cancer can be prevented, treated, and ultimately eliminated as a public health concern through primary, secondary, and tertiary prevention measures. These prevention strategies differ among females living without and with HIV as outlined in Fig 1, as per the WHO guidelines [6]. Primary prevention measures include HPV vaccination and educational interventions that create awareness related to cervical cancer risks and prevention and control strategies. Secondary prevention includes cervical cancer screening and treatment of precancerous lesions. Tertiary prevention includes the treatment of invasive cancer.

**Primary prevention.** Prophylactic HPV vaccination has been regarded as the most effective long-term strategy for preventing HPV infection, cervical cancer and other cancers associated with HPV [5]. The first HPV vaccine was licensed in 2006 [7]. HPV vaccines are designed to prevent over 95% of HPV infections caused by the common high-risk HPV types 16 and 18 as well as some cross protection against other less common HPV types that can cause cervical cancer [1]. The WHO recommends the following schedule: a one or two-dose schedule for girls aged 9–14 years, a one or two-dose schedule for girls and women aged 15–20 years and two doses with a 6-month interval for women older than 21 years [6]. A minimum of 2 doses and when feasible 3-doses remain necessary for those known to be immunocompromised and/or HIV-infected [6]. The high prevalence of HPV infections in HIV positive females emphasizes

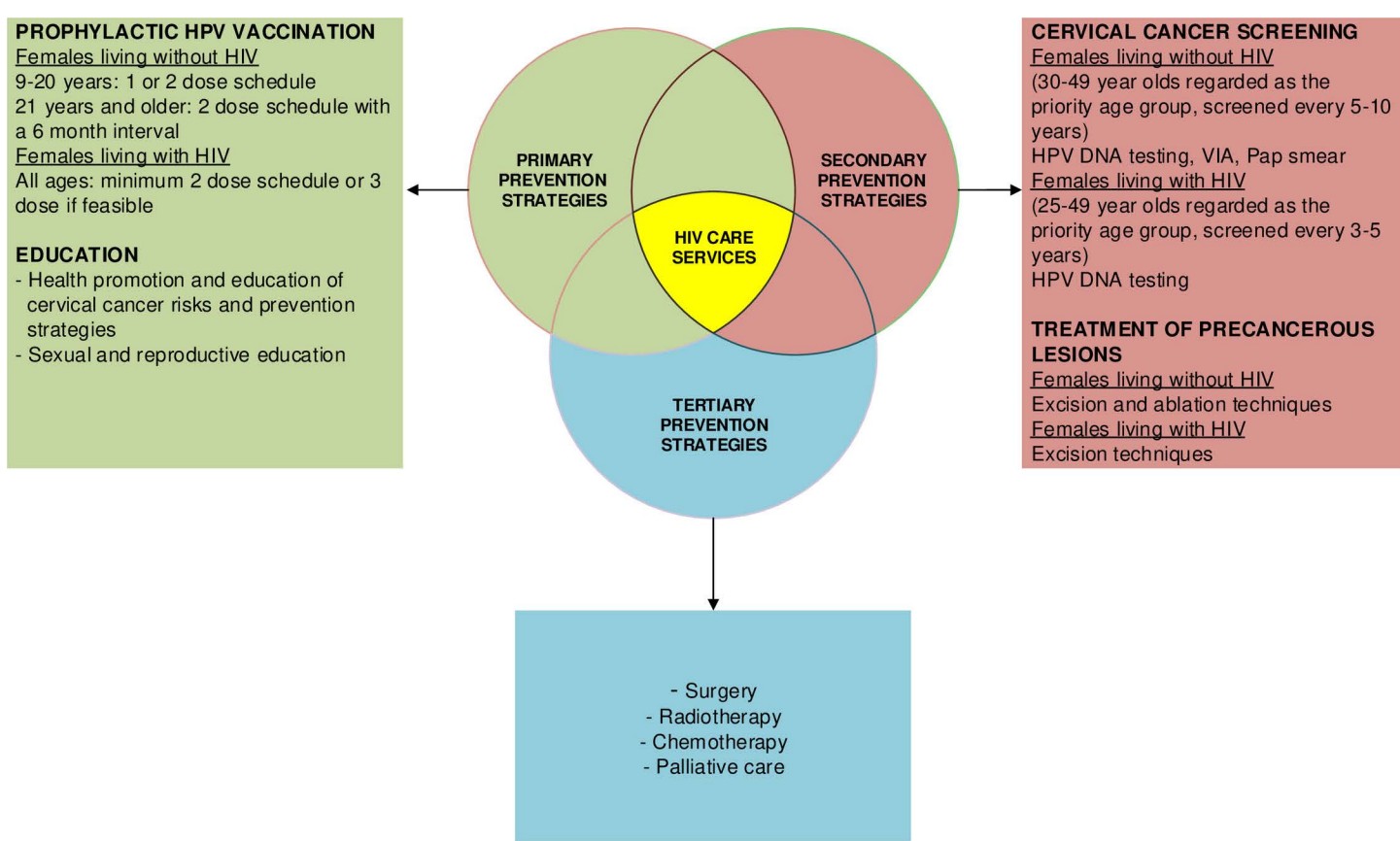

**Fig 1. Summary of cervical cancer prevention strategies as per WHO guidelines [ 6].**

how important it is to vaccinate this population. Furthermore, HPV vaccines are most effective if administered prior to HPV-exposure [7].

Educational interventions play a crucial role in creating awareness and imparting knowledge on the importance of cervical cancer prevention strategies and risks. In providing the necessary information females are empowered to make informed decisions about their health, seek HPV vaccination services and timely screening, as well as adopt proactive measures and ultimately reduce their risk of developing cervical cancer.

**Secondary prevention.** To prevent cervical cancer females can undergo various cervical screening tests to detect precancerous cells and thereafter receive timely and appropriate treatment to reduce the risk of progression to invasive cervical cancer. The conventional approach for screening involves cytology screening (conventional or liquid based), commonly known as Papanicoloau (Pap) test or Pap smear [8]. Advances in screening strategies include visual inspection with acetic acid (VIA) and more recently molecular tests such as high-risk HPV DNA testing [8].

In resource-limited settings, the predominant approach for treatment of cervical abnormalities is tissue ablation techniques such as cryotherapy or thermal ablation [9]. For women who are not eligible for ablation or in high income countries the main approach for treatment involves the excision of histologically confirmed cervical abnormalities [9].

The WHO recommends two approaches to screening and treatment namely, the "screen-and-treat" and the "screen, triage and treat" approach [6]. In the "screen-and-treat" approach treatment is administered to females solely based on a positive primary screening test (which means there is no secondary screening test and no histopathological diagnosis) [6]. In the "screen, triage and treat" approach the treatment decision is based on a positive result from the initial screening test, followed by a positive outcome in a subsequent test (referred to as a "triage" test), with or without histologically confirmed diagnosis [6]. For females in the general population, the WHO recommends HPV DNA detection as the screening method in a "screen-and-treat" or "screen, triage and treat" approach starting at the age of 30 years with regular screening every five to ten years [6]. For females living with HIV, it is recommended that HPV DNA detection be used in a "screen, triage and treat" approach starting at the age of 25 years with regular screening every three to five years [6].

Despite the development and implementation of cervical cancer prevention strategies this disease continues to be a public health concern. As a result of this, in 2020 the WHO introduced a "global strategy" to accelerate the elimination of cervical cancer [6]. This strategy proposes the following targets: 90% of adolescent girls globally should be vaccinated against HPV; 70% of women should undergo HPV screening and 90% of women diagnosed with cervical cancer should receive suitable follow-up treatment [6]. The aim of this global strategy is to reduce the incidence of cervical cancer to below a threshold of 4 cases per 100 000 women-years in every country [10]. Meeting these objectives set out by the WHO requires reconsideration of existing strategies to accelerate the adoption of HPV vaccination, cervical cancer screening and timely treatment of precancerous lesions, whilst optimizing rational resource allocation and use. The WHO advocates for the integration of these prevention strategies in other healthcare services to further enhance its effectiveness [11].

There has been a growing interest in integrating cervical cancer prevention strategies into HIV care programs [12]. For AGYW living with HIV, such integration can potentially improve access to HPV vaccination, and timely initiation of cervical cancer screening, resulting in early detection and treatment. Additionally, it may offer opportunities for the efficient utilisation of established healthcare resources and strengthen the health system's capacity to address both HIV and cervical cancer. Integrated healthcare approaches aim to leverage existing HIV care infrastructure to provide cervical cancer preventative services, which can encompass the full complement of prevention strategies. The integration not only streamlines healthcare delivery but also capitalizes on the regular contact that AGYW living with HIV have with healthcare providers. Supporting evidence highlighting the feasibility and outcomes of such integration programs has been published. For instance, Mwanahamuntu *et al*., [13] conducted a study on the integration of cervical cancer prevention services into HIV care services in Zambia and reported that over the course of 2.5 years more than 20 000 women had undergone cervical cancer screening following integration. Furthermore, studies conducted in Kenya, Mozambique and Botswana have reported cervical cancer screening within HIV care services to be feasible, acceptable, and effective [14–16].

Despite the potential benefits of integrating cervical cancer prevention and HIV care, there is still a limited understanding of its effectiveness in the real-world setting. As such, it is important to systematically assess the effectiveness of integrating such programs across various contexts.

This systematic review seeks to comprehensively assess existing evidence, by employing a mixed methods approach, to provide a holistic understanding of the outcomes and implications of integrating cervical cancer prevention strategies into HIV care programs. This review will delve into the uptake of HPV vaccination, initiation of cervical cancer screening, treatment of precancerous lesions, and educational interventions aimed at enhancing personal urgency and positive behavioural change. In addition, we will describe the knowledge, awareness, and willingness of AGYW living with HIV with regards to utilising and adhering to these strategies following integration into existing HIV care programs. Given the increasing global burden of cervical cancer, the continued burden of HIV, and the increased risk of HPV infections in AGYW living with HIV, this review will hold significant relevance and agency for public health initiatives aimed at improving the health and well-being of AGYW living with HIV as well as reducing the incidence and mortality rates of cervical cancer.

### Rationale for the review

Guided by the PICO (population, intervention, comparator and outcome) framework [17,18] the specific details of the population, intervention, comparator and outcomes for this review are outlined in Table 1.

**Population.** Focusing this review on AGYW (9–25 years) living with HIV is pertinent, given the distinct factors that heighten their lifetime vulnerability to cervical cancer and position them to gain significant benefits from integrated prevention strategies (Table 1). AGYW living with HIV are at a significantly higher risk of persistent HPV infection, leading to a more rapid progression to cervical precancer and potentially invasive cervical cancer [3]. This elevated risk necessitates early and sustained cervical cancer prevention efforts tailored specifically for this population to prevent early complications and premature death. Improving the uptake of these services in this age group would promote early prevention of cervical cancer with HPV vaccination and early detection or screening services thus ensuring significant long-term reduction in the incidence and mortality associated with cervical cancer.

**Table 1. Summary of the target population, interventions, comparator and outcomes (PICO) set out for this review.**

| PICO framework | Description |
|---|---|
| **Population (P)** | Adolescent girls and young women (AGYW) living with HIV (9–25 years) |
| **Interventions (I)** | Integration of cervical cancer prevention strategies with HIV care<br>Primary prevention:<br>1. HPV vaccination<br>2. Educational interventions<br>Secondary prevention:<br>1. Cervical cancer screening<br>2. Treatment of precancerous lesions. |
| **Comparator (C)** | Where studies are available that make use of before and after designs, we will explore the effectiveness of intervention versus no intervention or integration versus no integration. |
| **Outcomes (O)** | 1. A description of models used to integrate cervical cancer prevention strategies with existing HIV care programs will be identified and described.<br>2. The effectiveness of integrating cervical cancer prevention strategies into existing HIV care programs will be assessed in terms of the following outcomes:<br>  2.1 The uptake of cervical cancer prevention strategies (HPV vaccination, cervical cancer screening, treatment of precancerous lesions, and educational interventions) by AGYW living with HIV following integration.<br>  2.2 The knowledge, awareness, and willingness of AGYW living with HIV to utilise and adhere to cervical cancer prevention strategies following integration.<br>In addition, this review would also report qualitative data relating to the knowledge, awareness and willingness of AGYW living with HIV to utilise and adhere to cervical cancer prevention strategies following integration. |

**Intervention.** HPV vaccination is the most effective strategy for preventing cervical cancer with an efficacy rate of 95% [1,5]. Furthermore, HPV vaccines are most effective if administered prior to HPV-exposure and therefore primarily targeted at adolescent girls (9–14 years of age), prior to sexual debut [7]. Prioritising the uptake of HPV vaccination among AGYW living with HIV would contribute significantly towards reducing the global burden of cervical cancer globally. At present the WHO recommends initiating cervical cancer screening among females living with HIV at the age of 25 years as part of secondary prevention strategies [6]. Given that AGYW are a high-risk population [3], screening among this population would promote early detection and treatment of pre-cancerous lesions, thus reducing the likelihood of developing invasive cervical cancer later in life. Lastly, adolescence and young adulthood are generally regarded as the critical period for developing lifelong health behaviours. Increasing the uptake of educational interventions about HPV infection and cervical cancer among AGYW living with HIV could present an opportunity to increase knowledge and promote early adoption of cervical cancer prevention strategies among this population.

**Comparator.** The evidence base on cervical cancer prevention among AGYW and adult women within the general population is substantial and as such, this population will not be the focus of this review [19]. Given our specific interest in integrated services for AGYW living with HIV, we will explore utilisation of cervical cancer prevention services when integrated into HIV care pathways compared to no integration (Table 1).

**Outcomes.** The co-occurrence of cervical cancer and HIV represents a significant public health challenge and requires more effort to be placed on improving delivery and uptake of current cervical cancer prevention and control measures. Cervical cancer prevention strategies should encompass multidisciplinary approaches that include HPV vaccination, screening, treatment of precancerous lesions, and educational interventions (Table 1). Utilising existing HIV care programmes as a means of integrating cervical cancer prevention strategies into routine HIV care provides an effective way for improving cervical cancer prevention for the most at-risk population and should be prioritised. As mentioned, existing literature primarily focuses on isolated aspects of either HIV care or cervical cancer prevention, or some but not all cervical cancer preventative strategies, thus leaving a fragmented understanding of the holistic benefits of integrating services for these public health concerns. Previous studies have considered integration of these programs [12], although, more focus is placed on cervical cancer screening as a prevention strategy as opposed to a comprehensive approach that also includes vaccination, education, and treatment of precancerous lesions. Moreover, the search end dates of previous reviews are outdated - being more than six years old - and do not consider recent recommendations and advancements in the field [12]. This review aims to provide a comprehensive assessment of the effectiveness of, and knowledge, awareness, and willingness among AGYW living with HIV to utilise, integrated cervical cancer prevention strategies. The holistic approach intended for this review could provide further insights into the global stance on integration strategies. In doing this, gaps in terms of which strategies are underutilised, and less or more effective can be highlighted. Highlighting these gaps would be crucial in informing policy decisions on which prevention strategies require further development and planning. Limiting this review to just one strategy risks missing these crucial insights.

A mixed methods systematic review is an ideal research design for this study as it allows for an overall examination of both quantitative and qualitative data. The quantitative evidence will provide insights into the types of integration strategies adopted and their effectiveness, whilst the qualitative evidence can shed light on the knowledge, awareness, and willingness of AGYW living with HIV to utilise cervical cancer prevention strategies following integration. Findings from this review will have the potential to inform reforms to current policy and practice, thereby contributing to improving health outcomes for AGYW living with HIV.

## Research aims and objectives

### Aim

To describe the effectiveness of integrating cervical cancer prevention strategies into existing HIV care programs.

**Objectives**

1. To identify and describe integration models for cervical cancer prevention strategies and HIV care programs.

2. To assess the effectiveness of integrating cervical cancer prevention strategies into existing HIV care programs.

 2.1 To assess the uptake of HPV vaccination, cervical cancer screening, treatment of precancerous lesions, and educational interventions among AGYW living with HIV following integration into existing HIV care programs.

 2.2 To assess the knowledge, awareness, and willingness of AGYW living with HIV to utilise and adhere to cervical cancer prevention strategies following integration into existing HIV care programs.

## Methods

A comprehensive mixed methods review that evaluates primary studies employing qualitative, quantitative, and mixed methods will be conducted in alignment with the Joanna Briggs guidelines on mixed methods systematic reviews together with the Preferred Reporting Items for Systematic Reviews and Meta-Analyses (PRISMA) guidelines [20,21]. The protocol for this review was developed in accordance with the PRISMA-Protocol (PRISMA-P) guidelines and checklist (S1 Checklist. PRISMA-P 2015 checklist.) and is registered with the International Prospective Register of Systematic Reviews (PROSPERO) (ID: CRD42024535821), any changes to the published record will be reported [22]. The proposed timeline for this review is February 2024 to March 2025.

### Eligibility criteria

#### Inclusion criteria.

1. All primary research studies (including those with quantitative, qualitative, and mixed methods research designs) reporting findings on the effectiveness, knowledge, awareness, and willingness related to integrating cervical cancer prevention strategies (vaccination, education, screening, treatment of precancerous lesions, and educational interventions) into existing HIV care services will be considered. This includes all observational studies (cross-sectional and cohort), interventional studies (single-arm intervention studies, randomised control trials, cluster randomised control trials, cross-over trials, and non-randomised control trials) and qualitative studies (interviews, surveys, focus groups, ethnography, phenomenology, grounded theory studies and qualitative process evaluations).

2. The target population for this review are AGYW living with HIV, aged between 9 and 25 years of age. Therefore, only studies that include females within this age group living with HIV will be included.

3. Initially, we will not place any time limitations on our search strategy to ensure that we comprehensively scope the evidence base for relevant literature. We do anticipate that any literature relating to HPV vaccination in our target population will be published from 2006 onward in line with when the vaccine was first made available.

4. This systematic review has global significance that aims to address universal challenges affecting cervical cancer research on a global scale and therefore there will be no geographic limits placed on the search strategy.

5. We will not enforce any limitations on the language of publication. Instead, we will facilitate the translation of any potentially relevant publications into English to ensure their inclusion in the selection process and facilitate data extraction.

#### Exclusion criteria.

1. Studies that exclusively focus on AGYW living without HIV or that do not clearly specify the age and HIV status of the study participants will be excluded as this review focuses primarily on AGYW living with HIV.

2. Studies that include females younger the 9 or older than 25 years of age as this does not align with the target population for this review.

3. Studies that evaluate the integration of cervical cancer prevention strategies into programs other than HIV care services will not be considered.

4. This review will exclude all other review studies, modelling studies, evidence synthesis studies, case studies, case series, case reports and conference proceedings.

**Outcomes of interest**

1. We will identify and describe types or models of integrated services utilised by AGYW living with HIV to access cervical cancer prevention services. Sigfrid et al., 2017 identifies three models for integrating cervical cancer with HIV healthcare services [12]. The first model refers to within-clinic integration using internal staff, where the existing clinic structure and staff are used to incorporate a new set of services to complement services that are already provided [12]. In the second model, integration is achieved through co-location, where HIV and cervical cancer services are provided to the patient through coordination of care between different specialists or clinics within the same health care facility [12]. The third model involves complex programs of integration and coordination, including programs that integrated services by involving a range of different types of health workers (often from community health workers to clinical specialists) and facilities, and established systems to ensure clinical coordination and follow-up of patients [12]. Drawing on these three models, we will report on how cervical cancer prevention strategies are commonly integrated into HIV care services across the evidence base.

2. We will also report on the effectiveness of integrating cervical cancer prevention strategies into existing HIV care programs. This will be assessed in terms of the following outcomes:

   2.1 The uptake of cervical cancer prevention services (HPV vaccination, cervical cancer screening, treatment of precancerous lesions, and educational interventions) by AGYW living with HIV before versus after integration. Changes in measurable indicators related to the uptake of cervical cancer prevention strategies reported in included studies will be used to assess effectiveness. This can be numeric results (percentage or proportion), or measures of association related to uptake that have been reported before and after integration or those that compare uptake of services between integrated and non-integrated settings. To assess this outcome studies that employ a before-and-after study design will be used.

   2.2 The knowledge, awareness, and willingness of AGYW living with HIV to utilise and adhere to cervical cancer prevention strategies following integration. Changes in measurable indicators related to knowledge, awareness and willingness reported in included studies will be used to assess effectiveness. Again, this can include numeric results (percentage or proportion) or measures of association related knowledge, awareness and willingness to utilise and adhere to cervical cancer prevention services before and after integration or between integrated and non-integrated settings.

   In addition, this review will also report qualitative data (where available) relating to the knowledge, awareness and willingness of AGYW living with HIV to utilise and adhere to cervical cancer prevention strategies following integration.

**Search strategy**

A search strategy will be developed by one of the authors (KG) together with an information specialist (SN). A comprehensive literature search will be performed to enable capturing of as many relevant articles as possible based on the inclusion and exclusion criteria outlined. The following online electronic databases will be searched: PubMed, Cochrane Central

Library, EBSCO Host (Academic Search Premier, Africa-Wide Information, Cumulative Index to Nursing and Allied Health Literature [CINAHL], Health Source - Consumer Edition, Health Source: Nursing/Academic Edition, APA PsycArticles, APA PsycInfo), Web of Science, Scopus, and Google Scholar. Key words, medical subject headings (MeSH) and text words related to the themes; cervical cancer, HIV, prevention strategies and integration will be developed and then combined in the search strategy using Boolean operators, after which eligible articles will be identified. The search will be modified and applied to each electronic database (S1 Appendix. Database literature search strategy.). A standardized report template that will aid in keeping a record of all electronic databases searched, the search terms used and the total count of search results for each database will be developed and maintained. Reference lists of relevant studies will be searched for further articles in case they were missed during the primary searches. No geographical or language limitations will be applied.

## Study selection

All publications identified from the electronic database searches will be downloaded into a reference manager program, Zotero [23], and imported into a web-based platform called Rayyan for deduplication and screening [24]. Two authors (KG and NB) will independently screen articles by title and abstract and then full text articles screened for inclusion against the eligibility criteria. Reasons for exclusion of full text studies that do not meet the inclusion criteria will be recorded and reported in a table. Where the same study, using the same sample and methods, has been presented in different reports, we will collate these reports so that each study (rather than each report) is the unit of interest in our review to avoid over representation of datasets in each study in the systematic review results. Oversight of the study selection process will be provided by a third author (EA-D). Any disagreements that arise between the two authors (KG and NB) at any stage of the study selection process will be resolved through discussion and will involve a third author (EA-D) if necessary. A PRISMA flow diagram will be used to present the selection process and results of the search [21].

## Data extraction and management

Data extraction will be performed using piloted extraction forms by two authors (KG and NB) to ensure consistency across included studies. Standardised data extraction forms will be designed using Microsoft excel file (Version 2402) and used to record extracted data from included publications. Individual forms will be created and used to record quantitative and qualitative data (S2 Appendix. Data extraction form) extracted from quantitative, qualitative, and mixed method studies. A pilot data extraction process with a draft extraction form will be performed on approximately 10 articles to determine if all relevant information is being captured.

The Cochrane Handbook will be used to provide guidance for the inclusion of cluster randomised control trials in this review [25].

Two authors (KG and NB) will extract the data, and a third author (EA-D) will cross-check the data to ensure that all relevant data has been extracted. Any disagreements between KG and NB will be resolved by discussion. A third author (EA-D) will be involved to resolve any outstanding disagreement as necessary.

## Dealing with missing data

Study authors will be contacted via email regarding any unreported data or to seek clarification on study methods. Should the data remain unavailable, the data at hand will be analysed and the significance of any missing data will be discussed among authors (KG, NB and EA-D). Authors will then deliberate on the most suitable method for dealing with the missing data as per guidelines outlined in the Cochrane Handbook [26].

## Methodological quality assessment

All included studies (quantitative, qualitative, and mixed methods) will be assessed for methodological quality independently and in duplicate by two authors (KG and NB). Any disagreements will be resolved by discussion or by involving a third author (EA-D) if necessary.

**Quantitative studies.** To assess randomised controlled trials (RCTs) included in this review we plan to utilise the Cochrane risk of bias tool (ROB-2) [27]. This tool evaluates selection, performance, detection, attrition, reporting and additional sources of bias allowing us to categorize each of the included studies as having low, moderate, or high risk of bias. A summary of the assessment of each study with the overall judgement will be recorded and tabulated.

Non-randomised controlled trials (non RCTs) will be critically appraised using the Newcastle-Ottawa Scale (NOS) [28]. The NOS uses a star system for each included non RCT, which entails scoring stars based on a specific criterion. The overall quality of each study will be interpreted based on the total number of stars awarded. A comprehensive summary of all the assessments conducted, along with an overall judgement, will be recorded and tabulated.

**Qualitative studies.** The Critical Appraisal Skills Programme (CASP) quality assessment tool for qualitative studies will be applied to determine the rigour of qualitative methods used in included studies. The CASP tool will be used to examine the quality of a study in relation to 10 questions about research aims, appropriateness of methodology and design, recruitment strategy, data collection, researcher reflexivity, consideration of ethical issues, data analysis, statement of findings and the value of the research [29].

**Mixed methods studies.** The Mixed Methods Appraisal Tool (MMAT) will be applied to assess the risk of bias for mixed methods studies [30]. The following criteria will be used to assess the risk of bias:

1. Is there an adequate rationale for using a mixed methods design to address the research question?

2. Are the different components of the study effectively integrated to answer the research question?

3. Are the outputs of the integration of qualitative and quantitative components adequately addressed?

4. Are divergences and inconsistencies between quantitative and qualitative results adequately addressed?

5. Do the different components of the study adhere to the quality criteria of each tradition of the methods involved?

## Unit of analysis

The unit of analysis for the quantitative component of this review will be primarily at the individual level. Studies that will be selected for inclusion in this review will assess the effectiveness of integrating cervical cancer prevention strategies into HIV care services in improving the uptake of these strategies by AGYW living with HIV globally. Key epidemiological measures such as changes in the incidence, mortality, or prevalence of cervical cancer as well as utilisation of cervical cancer prevention services and rates thereof will be analysed at an individual level.

The unit of analysis for the qualitative component will be thematic. Qualitative data from included studies will be analysed to identify common themes related to the effectiveness of integrating cervical cancer prevention strategies into HIV care services. Themes related to the knowledge, awareness, and willingness of AGYW living with HIV to utilise these services following integration will be analysed at an individual level.

## Data synthesis

**Quantitative studies.** The findings from the included quantitative studies will be narratively summarised and graphically illustrated [31]. Study results will be expressed as either numeric results or measures of association, with their associated variation and confidence intervals. The magnitude of heterogeneity between the included studies will then be assessed quantitatively using the $I^2$ statistic [32]. $I^2$ values will be interpreted as follows: 0–40% might not be important; 30–60% may represent moderate heterogeneity; 50–90% may represent substantial heterogeneity and 75–100% represents considerable heterogeneity [32]. The significance of heterogeneity will be determined by the p-value as outlined in the Cochrane Handbook [32]. For studies with moderate to significant heterogeneity, a random effects model will be used to obtain a pooled estimate of the outcome. If heterogeneity is between 0–40% a random effects model

will also be used. In this instance, the random effects model will account for our lack of knowledge about why real, or apparent, intervention effects differ by considering the differences as if they were random [32].

Random effects models for meta-analysis will be performed using Review Manager software (RevMan 2020, V.5.4.1). If the heterogeneity detected is significantly high, a subgroup analysis will be performed to detect the possible sources [33]. A funnel plot will be used to assess for publication bias using R software (R Studio version 2023.03.0 + 386). Asymmetric distribution of the plot will indicate potential publication bias [34]. In addition, to statistically confirm whether the asymmetry is significant or not, the Begg and Egger's test will be performed using R software, where a p-value less than 0.05 indicates asymmetry and potential publication bias (R Studio version 2023.03.0 + 386) [34].

**Qualitative studies.** For the qualitative analysis thematic synthesis will be used to combine the findings of studies that describe the knowledge, awareness, and willingness of AGYW living with HIV to utilise and adhere to cervical cancer prevention strategies following integration [35]. The findings of included qualitative studies will be examined against the aims of this review, recurring patterns will be identified, and the qualitative data patterns will be interpreted by developing a coding framework. One article that closely answers the review objectives will be selected and used as a starting point to build a coding list. Two authors (KG and NB) will conduct "line-by-line" coding according to the content and meaning of the relevant findings of the article. The two authors (KG and NB) will subsequently discuss the "free" codes, with a third author (EA-D), develop an initial coding list, and independently test this list on two additional articles to determine if and how well the concepts translate from one study to another. The three authors (KG, NB and EA-D) will subsequently discuss the codes emerging from the data and agree on a preliminary coding framework. The remaining studies will then be coded line-by-line using the agreed coding framework, adding new codes as necessary. Relevant findings, reported anywhere in the primary qualitative studies, will be coded. If new codes arise during the analysis process, a discussion among the three authors will be conducted (KG, NB and EA-D) and the coding list will be amended accordingly. Two authors (KG and NB) will revisit articles already coded to determine if the new codes apply or not. This process will continue until they have extracted data from all the included articles.

Data extraction will be verified by a third author (EA-D). Review findings will then be synthesised from the data that have been given the same codes across the studies. Findings will be shared with the third author (EA-D) to review. Finally, we will re-read the included studies to check that we have extracted all data relevant to the findings.

**Combining quantitative and qualitative data.** Following the synthesis of quantitative and qualitative data independently, they will be combined using the methods and suggestions provided in the Cochrane Handbook [36]. According to the Johanna Briggs guidelines on data synthesis and integration for mixed method reviews, if the research question can be addressed by quantitative and qualitative research designs, a convergent integrated approach will be followed [37]. However, if the review aims to explore various aspects or dimensions of a particular phenomenon of interest the convergent segregated approach will be followed [37].

## Sensitivity analysis

To ensure the robustness and reliability of the findings of this review a sensitivity analysis will be conducted to assess the impact of the various methodological decisions on the results of this review and will be performed on both quantitative and qualitative components. The domains that will be considered include the quality of the included studies, sample size and the meta-analysis technique applied. If results remain consistent across the different analyses, the results can be considered robust as even with different decisions they remain the same/similar. If the results differ across sensitivity analyses, this is an indication that the results may need to be interpreted with caution [37].

## Assessment of quality of evidence

**Quantitative studies.** The quality of evidence for primary quantitative outcomes will be evaluated using the five Grading of Recommendations Assessment, Development and Evaluation (GRADE) criteria: risk of bias, consistency of effect, imprecision, indirectness, and publication bias [38]. To facilitate this process, we will utilize GRADEpro GDT

software and include footnotes to elucidate any determinations made to downgrade the quality of evidence. We will use the study design of each included study as a determining factor of whether to upgrade (i.e., observational studies) or downgrade (i.e. RCTs) the quality of evidence.

An assessment of each outcome will be presented in a GRADE Evidence Profile. Two authors (KG and NB) will detail the number of studies, the number of participants, and the numerical result of the meta-analysis for each outcome. The effects of interventions on the outcomes included in the GRADE Evidence Profiles will be interpreted according to magnitude of effect and certainty of the evidence, using GRADE guidance on informative statements to combine size and certainty of an effect [38]. If meta-analysis is unsuitable or units of analysis are incomparable, results will be presented in a narrative 'Summary of findings' table format, with a recognition of the imprecision in evidence due to the absence of a quantitative effect measure [39]. This process will be reviewed by a third author (EA-D).

**Qualitative studies.** To evaluate the confidence in synthesized qualitative findings, the Grades of Recommendation, Assessment, Development, and Evaluation- Confidence in the Evidence from Qualitative Reviews (CERQual) approach will be used [40]. This approach encompasses four key domains: methodological limitations, relevance of contributing studies to the research question, coherence of study findings, and adequacy of data supporting the study findings. For each outcome, two authors (KG and NB) will consolidate the findings from these four domains and offer rationale to elucidate any determinations made to downgrade the quality of evidence. This process will be reviewed by a third author (EA-D).

## Ethics and dissemination

Ethics approval will not be required for this review as the work constitutes a secondary analysis of published research, which is already available in the public domain. Findings from this systematic review will also serve as a guide for policy-makers in decision-making across countries with the dual burden of cervical cancer and HIV.

## Discussion

This review aims to address a critical gap in the existing literature by proposing a comprehensive mixed-methods systematic review on the effectiveness of integrating cervical cancer prevention strategies into HIV care programs. The dual burden of cervical cancer and HIV presents a significant public health challenge, especially in resource-limited settings, where both conditions are prevalent. There is a need to develop effective and integrated approaches that comprehensively address these dual health concerns simultaneously among AGYW living with HIV. The proposed research design, a mixed-methods systematic review, is well-suited to capture both quantitative effectiveness measures and qualitative insights into the knowledge, awareness, and willingness of AGYW living with HIV. By incorporating vaccination, screening, treatment of precancerous lesions and educational interventions into the assessment framework, this review aims to provide a holistic understanding of the potential benefits derived from the integration of prevention strategies. The strength of this review is the inclusion of current and up-to-date literature. Existing studies focus on isolated elements resulting in a fragmented understanding. The global search covering multiple databases enhances the robustness and reliability of this review.

## Conclusion

The anticipated outcomes of this systematic review could inform and improve implementation of current comprehensive cervical cancer prevention guidelines recommended by the WHO. Ultimately effective integration not only aligns with a holistic approach to women's health but also contributes substantially towards improving health outcomes for AGYW living with HIV. Reducing cervical cancer incidence and mortality can make a meaningful impact on global public health, thus emphasizing the importance of this systematic review.

## Research to practice

The review team anticipates that the findings from this proposed systematic review will enhance equitable access to cervical cancer prevention services, thereby promoting the quality of life for AGYW living with HIV.

## Supporting information

**S1 Checklist. PRISMA-P 2015 checklist.**
(DOCX)

**S1 Appendix. Database literature search strategy.**
(DOCX)

**S2 Appendix. Data extraction form.**
(DOCX)

## Author contributions

**Conceptualization:** Kimeshnee Govindsamy, Ntombifuthi Blose, Edina Amponsah-Dacosta.

**Data curation:** Kimeshnee Govindsamy, Susanne Noll.

**Formal analysis:** Kimeshnee Govindsamy, Ntombifuthi Blose, Edina Amponsah-Dacosta.

**Methodology:** Kimeshnee Govindsamy, Susanne Noll, Ntombifuthi Blose, Edina Amponsah-Dacosta.

**Project administration:** Kimeshnee Govindsamy.

**Resources:** Kimeshnee Govindsamy, Ntombifuthi Blose, Edina Amponsah-Dacosta.

**Software:** Kimeshnee Govindsamy, Ntombifuthi Blose, Edina Amponsah-Dacosta.

**Supervision:** Ntombifuthi Blose, Edina Amponsah-Dacosta.

**Validation:** Kimeshnee Govindsamy, Ntombifuthi Blose, Edina Amponsah-Dacosta.

**Visualization:** Kimeshnee Govindsamy, Ntombifuthi Blose, Edina Amponsah-Dacosta.

**Writing – original draft:** Kimeshnee Govindsamy.

**Writing – review & editing:** Susanne Noll, Ntombifuthi Blose, Edina Amponsah-Dacosta.

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
