## [Decision Letter · Decision Letter 0]

4 Dec 2024

Dear Dr. Govindsamy,

Dear authors, as you can see, the reviewers have requested substantial revisions to your manuscript. We are certainly willing to reconsider a revised submission, but please know that this is not preliminary acceptance of your paper. When returning your revised manuscript, please be sure to include a point-by-point summary of the suggestions of the reviewers that specifies how and where in the text you have addressed the suggestions.

We look forward to receiving your revised manuscript.

Kind regards,

Ricardo Ney Oliveira Cobucci, Ph.D

Academic Editor

PLOS ONE

Reviewers' comments:

Reviewer's Responses to Questions

**Comments to the Author**

1. Does the manuscript provide a valid rationale for the proposed study, with clearly identified and justified research questions?

Reviewer #1: Yes

Reviewer #2: Partly

2. Is the protocol technically sound and planned in a manner that will lead to a meaningful outcome and allow testing the stated hypotheses?

Reviewer #1: No

Reviewer #2: Partly

3. Is the methodology feasible and described in sufficient detail to allow the work to be replicable?

Reviewer #1: No

Reviewer #2: Yes

4. Have the authors described where all data underlying the findings will be made available when the study is complete?

Reviewer #1: No

Reviewer #2: No

5. Is the manuscript presented in an intelligible fashion and written in standard English?

Reviewer #1: Yes

Reviewer #2: Yes

You may also provide optional suggestions and comments to authors that they might find helpful in planning their study.

Reviewer #1: This systemic review protocol appears too ambitious in attempting to include all cervical cancer prevention strategies. Cervical cancer prevention encompasses primary prevention: HPV vaccination, and secondary prevention: various screening methods. Attempting to address all these strategies simultaneously within one systemic review could result in a lack of focus and clarity. Suggest narrow down to just one strategy.

It is recommended that the authors adopt the PICO framework to effectively structure this systematic review.

Firstly, the target population needs to be clearly defined. The authors should provide a rationale for selecting women aged 9-25 years, particularly given that screening methods is typically aimed at women aged 25-60.

Regarding the intervention or exposure focus, which is central to this study protocol, it should be clarified whether the primary emphasis is on HPV vaccination as a method of primary prevention or on screening methods. Including both aspects may lead to confusion among readers.

Additionally, the study protocol should specify the comparison or control group more explicitly.

The outcomes outlined in the study protocol are currently unclear and require further elaboration.

Reviewer #2: The authors describe the protocol of their proposed systematic review on the effectiveness of integrating cervical cancer prevention strategies into HIV care programs. This integration is vital for cervical cancer control in this vulnerable population and hence, the systematic review is well-timed.

However, there are certain major and few minor concerns regarding this protocol, as described below:

1. "Adolescent girls and young women (AGYW) face heightened risk of persistent HPV infection, a primary cause of cervical cancer." This statement is factually incorrect.

2. Why is the review being limited to studies reported after 2006. Though I agree that HPV vaccination was not available before 2006, but the review also includes cervical cancer screening and management of precancerous lesions. This restriction of time shall lead to erroneous results in these two areas of cervical cancer prevention.

3. "HPV is estimated to infect approximately 291 million women globally, with a significantly higher prevalence among women under the age of 25 years". Though this statement is correct, the authors should also mention that majority of these HPV infections are known to clear on their own.

4. "There are significant disparities in the prevalence of cervical cancer globally which highlights variations in accessibility..." This paragraph is not required and may be removed.

5. The sections on introduction of HPV vaccine, cervical cancer screening and treatment are very long and unnecessary. These need to be shortened extensively.

6. "According to WHO recommendations, females aged 9-20 years should receive one- or two-dose schedules and females 21 years and older should receive two-doses with a six- month interval". This statement is incorrect and in contradiction to the WHO Position Paper on HPV Vaccine, 2022. The reference provided for this statement is not accessible on the web. The authors need to check this information so that wrong information is not provided to the readers.

7. "Such integration can potentially improve access to cervical cancer screening, early detection, and pre-treatment services for AGYW living with HIV". Even in the WHO Guidelines, cervical cancer screening may begin at 21 years of age in women living with HIV. Hence, limiting this review to studies conducted in adolescent girls and young women would greatly limit the results reported thereof. For instance, the study of Mwanahamuntu et al reported cervical cancer screening in women living with HIV with mean age of 34 years.

8. "Moreover, the search end dates of these reviews are outdated being more than six years old- and do not consider recent recommendations and advancements in the field." Which reviews are being referred to here? No references have been provided.

9. For Objective 2, what is the definition of effectiveness of integrating cervical cancer prevention strategies into HIV care programs? Which parameters would be extracted from the included studies for this outcome?

10. How would knowledge, awareness, and willingness combined in the review? The reporting units of various studies could vary significantly.

11. What would be the comparator for the various parameters related to integration of cervical cancer prevention strategies in HIV care programs?

12. For Outcome 1, what would be the data extracted from the included studies? How would that be assessed in the review?

13. For Outcome 2, there should be a comparator to assess the effectiveness against non-integration. In absence of a comparator, the results and conclusion would be less meaningful.

**Do you want your identity to be public for this peer review?** For information about this choice, including consent withdrawal, please see our Privacy Policy

Reviewer #1: No

Reviewer #2: No

---

## [Author Response · Author response to Decision Letter 1]

18 Jan 2025

Dear Academic Editor and Reviewers

Please find responses to reviewer's comments in uploaded document named "Response to reviewer"

---

## [Decision Letter · Decision Letter 1]

18 Feb 2025

Dear Dr. Govindsamy,

Thank you for submitting your manuscript to PLOS ONE. After careful consideration, we feel that it has merit but does not fully meet PLOS ONE’s publication criteria as it currently stands. Therefore, we invite you to submit a revised version of the manuscript that addresses the points raised during the review process.

Dear, 

a reviewer has also asked for revisions to the manuscript that are important in order to improve the quality and that the reviewer and the editor can consider publishing in the new revision. Authors should carefully evaluate these suggestions, resolve all of them in the manuscript if they consider it possible, or reply to the reviewer with an acceptable justification as to why they do not agree, or are unable to revise as suggested.

We look forward to receiving your revised manuscript.

Kind regards,

Ricardo Ney Oliveira Cobucci, Ph.D

Academic Editor

PLOS ONE

Journal Requirements:

Reviewers' comments:

Reviewer's Responses to Questions

**Comments to the Author**

1. Does the manuscript provide a valid rationale for the proposed study, with clearly identified and justified research questions?

Reviewer #1: Yes

Reviewer #2: Partly

2. Is the protocol technically sound and planned in a manner that will lead to a meaningful outcome and allow testing the stated hypotheses?

Reviewer #1: Yes

Reviewer #2: Partly

3. Is the methodology feasible and described in sufficient detail to allow the work to be replicable?

Reviewer #1: Yes

Reviewer #2: No

4. Have the authors described where all data underlying the findings will be made available when the study is complete?

Reviewer #1: Yes

Reviewer #2: Yes

5. Is the manuscript presented in an intelligible fashion and written in standard English?

Reviewer #1: Yes

Reviewer #2: Yes

You may also provide optional suggestions and comments to authors that they might find helpful in planning their study.

Reviewer #1: The authors have positively responded to the suggestions and have made significant edits, greatly improving this study protocol. I have no further issues.

Reviewer #2: 1. The rationale of including AGYW living with HIV is still not acceptable. The systematic review protocol aims to include all cervical cancer prevention strategies. When the WHO guidelines advocate cervical cancer screening in WLHIV starting at 25 years of age, how would the data align with the objective of the usefulness of cervical cancer screening in HIV care? The authors need to rethink the age group to be included. It would be better to have two age tiers.

2. The rationale of including studies post-2006 is also not acceptable. On the one hand, the authors claim that their systematic review shall have a global significance and on the other, they state that pre-2006, studies on cytology-based screening methods do not align with the WHO global guidelines. The authors need to revisit the WHO cervical cancer screening guidelines. The guidelines state that primary HPV-based screening is the standard. However, if this is not feasible, standardized cytology-based cervical cancer screening is acceptable. Hence, inclusion of only studies published after 2006 would bias the results highly and is not advocated. Duplication of earlier reviews is never an issue with a new systematic review and studies are not excluded solely on this basis.

3. In order to report on the effectiveness of integration of cervical cancer prevention strategies into HIV care programs, only studies with a before-and-after design should be included. This should be specifically stated.

**Do you want your identity to be public for this peer review?** For information about this choice, including consent withdrawal, please see our Privacy Policy

Reviewer #1: No

Reviewer #2: No

---

## [Author Response · Author response to Decision Letter 2]

3 Apr 2025

Response to Reviewer 1

Comment 1:

The rationale of including AGYW living with HIV is still not acceptable. The systematic review protocol aims to include all cervical cancer prevention strategies. When the WHO guidelines advocate cervical cancer screening in WLHIV starting at 25 years of age, how would the data align with the objective of the usefulness of cervical cancer screening in HIV care? The authors need to rethink the age group to be included. It would be better to have two age tiers.

Author’s response to comment 1:

We would like to thank the reviewer for this insightful concern regarding the age group for the participants in this review. While we acknowledge the WHO guidelines recommending cervical cancer screening in females living with HIV starting at the age of 25 years, we firmly believe in the merit this review would have if we were able to successfully demonstrate the effectiveness of screening in AGYW living with HIV. This would aid in facilitating timely identification of HPV infection before progression to cervical cancer.

As mentioned in the rationale AGYW living with HIV experience a higher risk of persistent infection and rapid progression to cervical precancerous lesions. Studies have shown that females living with HIV have a higher prevalence of high-risk HPV types even before the recommended age of 25 years [1]. For example, in South Africa HIV infection was found to be linked to persistence of HPV infection and a higher risk of Pap smear abnormalities in South African females aged 17–21 years [2]. Furthermore, cervical cancer rates begin to increase from 25-29 years [3]. Early screening in AGYW living with HIV could therefore serve as a preventive strategy in detecting and treating precancerous lesions before progression to cervical cancer in older age groups. If this review can demonstrate integration of the prevention strategies with HIV care to be effective among AGYW living with HIV this would mean that females in this population would make use of these integrated services and in turn potentially reduce the cervical cancer rates in females older than 25 years.

Comment 2:

The rationale of including studies post-2006 is also not acceptable. On the one hand, the authors claim that their systematic review shall have a global significance and on the other, they state that pre-2006, studies on cytology-based screening methods do not align with the WHO global guidelines. The authors need to revisit the WHO cervical cancer screening guidelines. The guidelines state that primary HPV-based screening is the standard. However, if this is not feasible, standardized cytology-based cervical cancer screening is acceptable. Hence, inclusion of only studies published after 2006 would bias the results highly and is not advocated. Duplication of earlier reviews is never an issue with a new systematic review and studies are not excluded solely on this basis.

Author’s response to comment 2:

We would like to thank the reviewer for this insightful comment on the timeline for this systematic review.

We have now revised the methods section for this protocol with the guidance provided which can be found on page 15, line 311-314, and reads as follows:

“Initially, we will not place any time limitations on our search strategy to ensure that we comprehensively scope the evidence base for relevant literature. We do anticipate that any literature relating to HPV vaccination in our target population will be published from 2006 onwards in line with when the vaccine was first made available.”

Comment 3:

In order to report on the effectiveness of integration of cervical cancer prevention strategies into HIV care programs, only studies with a before-and-after design should be included. This should be specifically stated.

Author’s response to comment 3:

We would like to thank the reviewer for this comment. We have included a statement in the protocol to highlight this point.

We have revised the outcome outlined for this review which can be found on page 16-17, line 350-362, which now reads:

2. “We will also report on the effectiveness of integrating cervical cancer prevention strategies into existing HIV care programs. This will be assessed in terms of the following outcomes:

2.1 The uptake of cervical cancer prevention services (HPV vaccination, cervical cancer screening, treatment of precancerous lesions, and educational interventions) by AGYW living with HIV before versus after integration. Changes in measurable indicators related to the uptake of cervical cancer prevention strategies reported in included studies will be used to assess effectiveness. This can be numeric results (percentage or proportion), or measures of association related to uptake that have been reported before and after integration or those that compare uptake of services between integrated and non-integrated settings. To assess this outcome studies that employ a before-and-after study design will be used.”

References:

1. Burmeister CA, Khan SF, Schäfer G, Mbatani N, Adams T, Moodley J, et al. Cervical cancer therapies: Current challenges and future perspectives. Tumour Virus Res. 2022;13: 200238. doi:10.1016/j.tvr.2022.200238

2. Whitworth H, Changalucha J, Baisley K, Watson‐Jones D. Adolescent Health Series: HPV infection and vaccination in sub‐Saharan Africa: 10 years of research in Tanzanian female adolescents ‐ narrative review. Trop Med Int Health. 2021;26: 1345–1355. doi:10.1111/tmi.13660

3. Moscicki A-B, Perkins RB, Saville M, Brotherton JML. Should Cervical Cancer Screening be Performed Before the Age of 25 Years? J Low Genit Tract Dis. 2018;22: 348–351. doi:10.1097/LGT.0000000000000434

---

## [Decision Letter · Decision Letter 2]

17 Apr 2025

Effectiveness of integrating cervical cancer prevention strategies into HIV care programmes: A mixed-methods systematic review protocol.

PONE-D-24-24785R2

Dear Dr. Govindsamy,

We’re pleased to inform you that your manuscript has been judged scientifically suitable for publication and will be formally accepted for publication once it meets all outstanding technical requirements.

Kind regards,

Ricardo Ney Oliveira Cobucci, Ph.D

Academic Editor

PLOS ONE

Additional Editor Comments (optional):

The editor considers that the corrections made by the authors in the protocol were sufficient for the manuscript to be accepted, as recommended by one of the reviewers and after a careful editorial evaluation. 

Reviewers' comments:

Reviewer's Responses to Questions

**Comments to the Author**

1. Does the manuscript provide a valid rationale for the proposed study, with clearly identified and justified research questions?

Reviewer #2: Partly

2. Is the protocol technically sound and planned in a manner that will lead to a meaningful outcome and allow testing the stated hypotheses?

Reviewer #2: Yes

3. Is the methodology feasible and described in sufficient detail to allow the work to be replicable?

Reviewer #2: Yes

4. Have the authors described where all data underlying the findings will be made available when the study is complete?

Reviewer #2: Yes

5. Is the manuscript presented in an intelligible fashion and written in standard English?

Reviewer #2: Yes

You may also provide optional suggestions and comments to authors that they might find helpful in planning their study.

Reviewer #2: I am not convinced about the rationale of restricting this review to AGYW living with HIV. Though this population may be at a higher risk of persistent HPV infection, the review should be more encompassing to justify the aim of utility of integrating cervical cancer prevention strategies with HIV care.

**Do you want your identity to be public for this peer review?** For information about this choice, including consent withdrawal, please see our Privacy Policy

Reviewer #2: No

---

## [Editor Report · Acceptance letter]

PONE-D-24-24785R2

PLOS ONE

Dear Dr. Govindsamy,

I'm pleased to inform you that your manuscript has been deemed suitable for publication in PLOS ONE. Congratulations! Your manuscript is now being handed over to our production team.

Kind regards,

on behalf of

PROFESSOR Ricardo Ney Oliveira Cobucci

Academic Editor

PLOS ONE